
# Dynamical forcings in heavy precipitation events over Italy: lessons from the HyMeX SOP1 campaign

M. Marcello Miglietta[1], Silvio Davolio[2]

[1]CNR-ISAC, Padua, 35127, Italy

5 [2]CNR-ISAC, Bologna, 40129, Italy

*Correspondence to*: M. Marcello Miglietta  (m.miglietta@isac.cnr.it)

**Abstract.** The first Special Observation Period (SOP1) of HyMeX (Hydrological cycle in the Mediterranean eXperiment) was held in Fall 2012 and focused on heavy precipitation events (HPEs) and floods in the northwestern Mediterranean. Nine intensive observation periods (IOPs) involved the three Italian target areas (north-eastern Italy, NEI; Liguria and Tuscany, 10  LT; central Italy, CI), enabling an unprecedented analysis of precipitation systems in these regions. In the present work, we highlight the major findings emerging from the HyMeX campaign and in the subsequent research activity over the three target areas, by means of conceptual models and through the identification of the relevant recursive mesoscale features.

For NEI, two categories of events (Upstream and Alpine HPEs) have been identified, which differ mainly in the temporal evolution of the stability of the upstream environment and of the intensity of the impinging flow (i.e., the Froude number). 15  The numerical simulation of convection in the Po Valley was found very sensitive to small changes in the environmental conditions, especially when they are close to the threshold between "flow-over" and "flow-around" regimes. Some mesoscale features (e.g., the presence of a shallow pressure minimum in the eastern Po Valley) were identified as fundamental to adequately simulate the detailed evolution of severe convective episodes.

For LT, HyMeX SOP1 focused on orographically-enhanced precipitation over the Apennines and quasi-stationary mesoscale 20  convective systems over the sea or close to the coast. For the latter category of events, associated with the majority of the recent HPEs in the area, local-scale or large-scale convergence lines appear fundamental to trigger and sustain convection. These lines are affected not only by the orography of the region, but also by perturbations induced by Sardinia and Corsica on the environmental flow. Cold pools formed via evaporation of precipitation also played a major role in determining the position of the trigger at later times. The accurate representation of the moisture structure below 2 km is the key to an 25  accurate simulation of the timing and location of precipitation.

For CI, a high low-level moisture content and marked low-level convergence over the sea were critical to support deep convection in IOPs affecting the Tyrrhenian coast. Also, an elevated moisture plume from the Tropics was observed to locally reinforce the intensity of the updrafts. For HPEs affecting the Adriatic regions, generally a cut-off low over the Tyrrhenian Sea induces intense Bora over the Adriatic basin. Low-level convergence triggers convection over the sea, while



orographic uplift produces stratiform precipitation. The Adriatic Sea plays a critical role mainly through air-sea exchanges, which modify the characteristics of the flow and in turn the effect of the orographic forcing.

## 1 Introduction

The peculiar characteristics of the Mediterranean region, a nearly enclosed basin surrounded by complex terrain close to the coast, makes the area particularly prone to natural hazards related to the water cycle. Intense events, such as heavy rainfall

and floods, still pose a significant threat to people (Llasat et al., 2013), in spite of the noteworthy improvements in forecasting, emergency management, and defensive measures. Hence, the interest in improving the understanding and forecasting of severe weather events is clear for both scientific research and operational activities.

The Hydrological cycle in the Mediterranean eXperiment (HyMeX, http://www.hymex.org; Drobinski et al., 2014) was an international program devoted to advance the scientific knowledge of the water cycle in the Mediterranean basin. This goal

was pursued through the monitoring, analysis and modeling of the regional hydro-meteorological cycle in a seamless approach. Phenomena were investigated at different temporal and spatial scales, ranging from the inter-annual/decadal variability of the atmosphere–land–ocean coupled system to single case studies of severe weather. The experimental activity, a key component of HyMeX, was based on atmospheric, oceanic and hydrological monitoring covering a period of ten years, from 2010 to 2020. Within this time frame, Special Observation Periods (SOPs), shorter periods of intensive monitoring,

were planned. In particular, the first Special Observation Period (SOP1), between 5 September and 6 November 2012, focused on heavy precipitation events (HPEs) and floods in the Western Mediterranean (WMED). During SOP1, twenty Intensive Observation Periods (IOPs) were undertaken (Ducrocq et al., 2014), nine of which involved the Italian regions (see Tab. 5 in Ferretti et al., 2014). The second field campaign, named SOP2, took place in winter 2013 (February–March) and was dedicated to the study of intense air–sea exchanges and dense water formation. HyMeX thoroughly investigated the

processes responsible for HPEs, including the origin of moist air masses in pre-convective conditions (Duffourg and Ducrocq, 2011, 2013; Duffourg et al., 2018; Flaounas et al., 2019; Lee et al., 2019) and the link between these air masses and intense rainfall. During SOP1, dedicated observing platforms were managed with the objective of documenting the connection between moist flows and HPEs over the coastal areas of WMED (Ducrocq et al., 2014; Bock et al., 2016; Lee et al., 2017; Khodayar et al., 2018).

The need of progress in the monitoring and prediction of these events, with the purpose of preventing or reducing societal losses, represented the strong motivation that induced a large and active participation of the Italian community in SOP1 (Ferretti et al., 2014; Davolio et al., 2015a). The activities carried out during the field campaign and planned for the upcoming years represented a unique opportunity for the study of intense orographic precipitation, exploiting the synergy between observations and available model simulations. The Italian peninsula, extending from the southern Alps in the north

to the central Mediterranean Sea in the south, is particularly exposed to natural hazards associated with the water cycle and to the consequent hydro-geological effects, often amplified by its complex morphology; thus, the scientific aims of HyMeX





were particularly relevant for the Italian territory. The steep slopes of the Alps and the Apennines in the proximity of the Mediterranean, and the Mediterranean Sea itself, which acts as a source of moisture and heat, are key factors in determining the convergence and the uplift of moist and unstable air, which are responsible for triggering convection over Italy and

surrounding seas. Moreover, HPEs persisting for several hours within the small and densely urbanized watersheds with steep slopes, which characterize the Italian area, can generate devastating floods in a relatively short time (Miglietta and Regano, 2008; Mastrangelo et al., 2011; Silvestro et al., 2012; Vulpiani et al., 2012; Buzzi et al., 2014; Gascon et al., 2016; Fiori et al., 2017). The national socio-economic impact of these kinds of events is relevant, as indicated by the high number of casualties and damages reported in recent years (e.g., Guzzetti et al., 2005; Salvati et al., 2010).

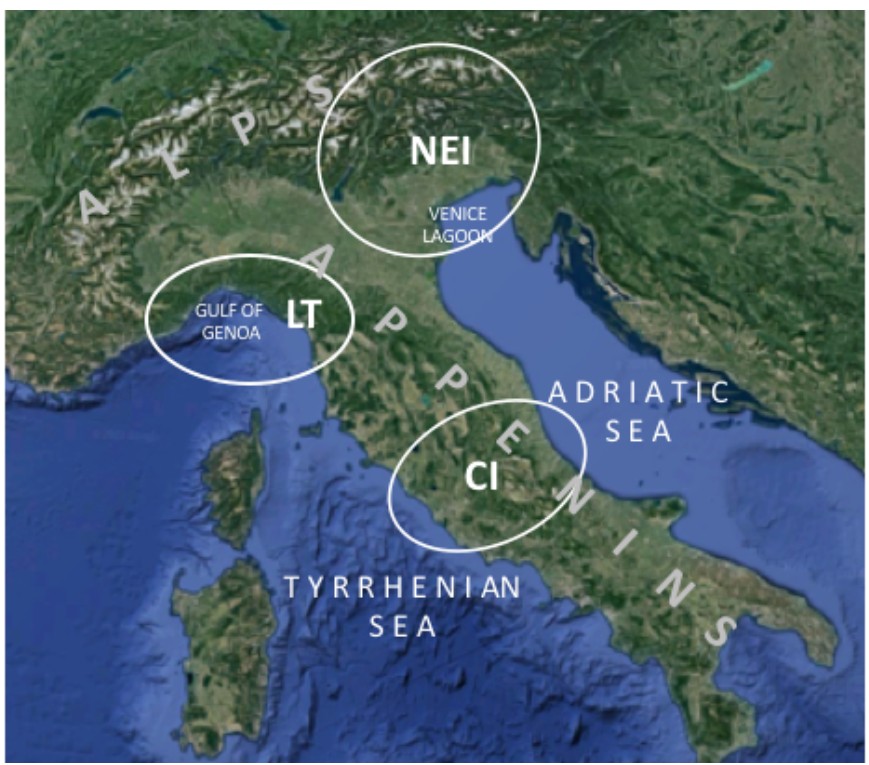


Figure 1: Italian target areas and geographic places mentioned in the text (Figure background from: Imagery ©2021 Landsat / Copernicus, Data SIO, NOAA, U.S. Navy, NGA, GEBCO, Imagery ©2021 TerraMetrics, Map data ©2021 GeoBasis-DE/BKG (©2009), Google, Inst. Geogr. Nacional).

Among the target areas in the WMED basin monitored during the field campaign, three were selected in Italy (Fig. 1): an Alpine/pre-Alpine site in northeastern Italy (NEI), characterized by the maximum amount of mean annual precipitation (Isotta et al., 2014), two target areas in regions frequently affected by HPEs, such as Liguria and Tuscany (LT; e.g., Rebora et al., 2013), and central Italy (CI; Ferretti et al., 2012). The extraordinary deployment of advanced instrumentation, including aircrafts, and the availability of several weather, hydrological and ocean models operated in real-time, allowed an

unprecedented monitoring and analysis of HPEs in Italy. The activity stimulated the collaboration in the Italian



meteorological community and reduced its traditional fragmentation. Davolio et al. (2015a) described the remarkable organizational efforts and daily activities at the HyMeX national Virtual Operational Centre (VOC), hosted by the Center of Excellence in Telesensing of Environment and Model Prediction of Severe events (CETEMPS) in L'Aquila, during the SOP1: scientists and forecasters from fourteen research and operational Italian institutions moved to the VOC on weekly shifts,

promoting a fruitful collaboration. The VOC supported the main HyMeX Operational Centre (HOC) in Montpellier (France), providing, analyzing and discussing numerical weather prediction (NWP) model outputs and nearly real-time observations during the international briefings, thus contributing to the success of the field campaign. On the other hand, Ferretti et al. (2014) provided a scientific overview of some events that affected the Italian area during SOP1, investigating in detail the response of the different operational modeling chains specifically for three case studies (IOP2, IOP13 and IOP19),

characterized initially by convection over the sea and orographic precipitation at later times.

During SOP1, negative or slightly positive values of the North Atlantic Oscillation (NAO) index prevailed, favoring cyclonic circulation and making the environmental conditions prone to precipitation systems crossing southern Europe. Several troughs entered the WMED and reached the Italian regions after having affected other target areas, over Spain or France, often producing westerly–to–southwesterly flow over Italy (IOP2, IOP6, IOP7b, IOP12, IOP19); occasionally, mesoscale

cyclones developed (IOP4, IOP16a) and only in few events lee cyclogenesis occurred over the Gulf of Genoa or a deep trough formed over the Tyrrhenian Sea (IOP13, IOP16c, IOP18). In general, the monthly precipitation amounts were close to the climatological values in September, but well above in October (Khodayar et al., 2016); the most affected sites were the Cévenne-Vivarais (CV), the French Southern Alps, as well as the LT target area in Italy.

The purpose of the present paper is to summarize what we know about the mechanisms responsible for HPEs in the three

Italian target areas, highlighting the main findings emerging from the HyMeX campaign and the subsequent scientific activity, using also conceptual models to summarize the main factors responsible for these events. In these effort, Section 2 provides a general overview of our knowledge on HPEs in the WMED region, while Section 3 is dedicated to the main results for the three Italian target areas (NEI, LT, CEI). Conclusions are drawn in Section 4.

## 2 An overview of western Mediterranean HPEs

The synoptic settings conducive to HPE in the WMED are relatively well known from past international field campaigns, such as the Mesoscale Alpine Programme (MAP, Bougeault et al., 2001), the Medex project (Jansa et al., 2014), and subsequent studies (Rotunno and Houze, 2007; Nuissier et al, 2008; Winschall et al., 2012; Grazzini et al., 2019); in some areas, such as the Alps, they are often characterized by higher-than-average predictability (e.g. Grazzini, 2007), thus they are adequately simulated at least at short leading time. HPEs in the northern coasts of WMED often develop eastward of an

upper-level trough (Nuissier et al., 2011) deepening over the basin, which is responsible for steering warm and moist low-level unstable flow from the sea (characterized by high values of equivalent potential temperature and precipitable water) towards the mountainous coasts of the region. Lee cyclogenesis (Buzzi et al., 2003), frontal interaction with the Alps (Buzzi



and Alberoni, 1992), low-level flow modifications due to the shape of the mountain barrier and to humidity/stability gradients (Rotunno and Ferretti, 2001), moisture supply by air masses flowing over the western Mediterranean (Buzzi and Foschini, 2000) represent mechanisms contributing to HPEs, which have received particular attention so far. Furthermore, Jansa et al. (2001) indicated that about 90% of the HPEs in the WMED involve the presence of a cyclonic center in their vicinity, usually in locations favorable to the creation and intensification of a low-level flow favoring convective initiation.

However, a lot of uncertainty is still associated with the description and accurate simulation of mesoscale processes and mechanisms conducive to initiation and intensification of precipitation systems, especially when convection dominates. Hence, the HyMeX field campaign was entirely devoted to investigate the small scales of convection down to microphysics, and to provide new evidences and crucial insights on the genesis and evolution of quasi-stationary mesoscale convective systems (MCS) in a complex topographic environment.

Idealized studies of moist flow interacting with orography (e.g., Miglietta and Buzzi, 2001, 2004; Miglietta and Rotunno, 2009, 2010; Davolio et al., 2009; Bresson et al., 2012; Kirshbaum et al., 2019) represent the cornerstone and the reference for the interpretation of new results. They have shown that the environmental conditions, including the characteristics of the upstream flow, may affect the mechanisms that trigger convection, such as orographic lifting, low-level wind convergence and cold pools formation (Ducrocq et al., 2008, 2016).

When HPEs are observed near the mountains or at some distance upstream, deep convection is likely triggered by the direct orographic uplift or by the deflection or blocking action exerted by the orography, which renew convection triggering at the same location (Davolio et al., 2006). As long as the same environment conditions persist, a back-building process may operate: new convective cells are repeatedly triggered at the same location, while older cells are transported downstream by the mid-to-upper level steering flow (Chen and Lin, 2005; Miglietta and Rotunno, 2014). This topic received a lot of benefits from the HyMeX research outcomes, as further detailed in the following. Figure 2 from Ducrocq et al. (2016) shows the typical configuration for MCS development in southern France, but similar conditions are associated to HPEs in the other regions of the northern side of the WMED, such as the LT target area or the north Adriatic region.



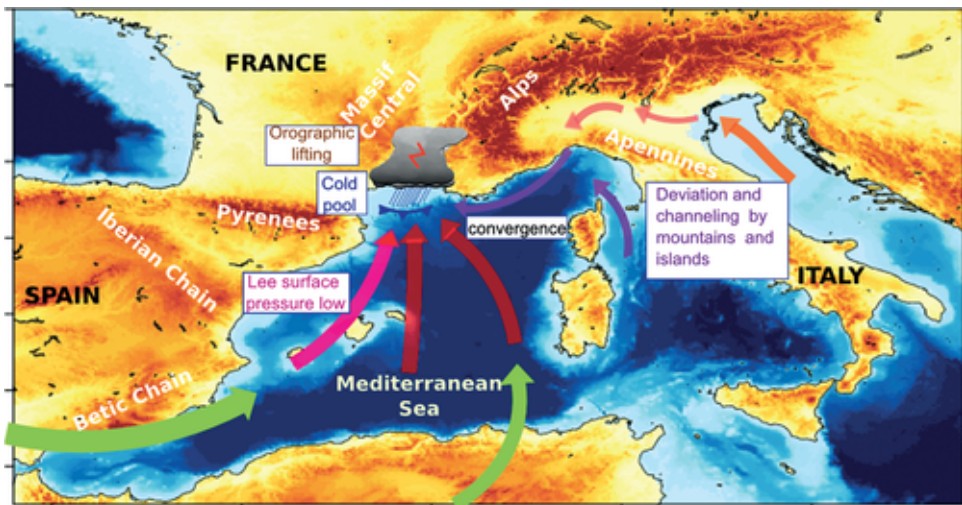

Figure 2: Schematic of the main low-level mechanisms responsible for a sample case of HPE in the western Mediterranean region together with geographical locations (From Ducrocq et al., 2016).


Convective initiation may also occur over the sea and over the plains. In this case, mechanisms of convection triggering and sustainment other than direct orographic uplift may occur, such as persistent low-level convergence induced by the alteration of the low-level flow by mountains and islands. In more details, convergence can be associated with: flow blocking and deflection, or channeling by the complex orography surrounding the basin, which also constraints the moisture transport in

the area (Bresson et al., 2012; Buzzi et al., 2014; Barthlott and Davolio, 2016, Scheffknecht et al., 2016); the presence of an even shallow cyclonic circulation (Duffourg et al., 2016; Khodayar et al., 2016); frontal passage (Lee et al., 2016); interaction between sea breezes and drainage winds induced by mountainous islands, like Corsica or Sardinia (Barthlott and Kirshbaum, 2013). Often, it is the interplay of different factors that allows the development of MCSs (Lee et al., 2017). Occasionally, the transport of water vapor within thermally driven circulations, such as valley winds, slope winds and sea

breezes, may cause spatial inhomogeneities in the pre-convective conditions and in the development of isolated deep convection (Adler et al., 2016).

In any case, a low-level evaporative cold pool, whose intensity is modulated by the moisture vertical distribution (Lee et al., 2016, 2018), may form under the MCS and behaves as an obstacle (Duffourg et al., 2016; Bouin et al., 2017) that lifts the low-level flow at its leading edge or locally modifies the low-level circulation, even enhancing flow convergence and

upward motion (Duffourg et al., 2016). Cold pool may remain stationary when counteracted by the environmental flow (Miglietta and Rotunno, 2012; Davolio et al., 2016), generating quasi-stationary MCSs. Moreover, once the convective system is formed, it may interact with the mountains and islands of the region, and the cold pool may be blocked or may even push the MCS upstream, away from the mountain barrier (Duffourg et al., 2018). This cold-pool convection-triggering mechanism occurs more frequently when the ambient flow is relatively dry or weak (Bresson et al., 2012). In these

conditions, the accurate rainfall prediction requires a detailed description of the low-level vertical moisture distribution (Lee





et al., 2018): the variability and stratification of atmospheric moisture in the lower troposphere determines when, where, and how intense convection will develop and if a cold pool may form (Lee et al., 2016).

In many cases, multicell V-shaped MCS have been observed (Davolio et al., 2009; Rebora et al., 2013; Lee et al., 2016), often forming just offshore. These systems are characterized by retrograde regeneration ("back-building" process – e.g. 165 Schumacher and Johnson, 2005), since convective cells are repeatedly generated upstream while older cells move downstream. Thus, they appear anchored over the sea, but intense convective cells and precipitation affect the same region also inland (Duffourg et al., 2016, Buzzi et al., 2014; Fiori et al., 2017). The complex multi-scale interaction between the moist ambient inflow extracting moisture and heat from the sea surface, deep convection and the topography makes difficult the precise prediction of the timing, location, and amount of precipitation associated with these systems.

Determinant for the development of precipitating convection in the WMED is the presence of moisture, which may originate from either remote or local sources (Ricard et al., 2012; Krichak et al., 2015; Khodayar et al., 2016). Depending on the synoptic conditions, the Mediterranean Sea may account for the majority of humidity (Duffourg and Ducrocq, 2011): this is the case when an anticyclonic flow dominates during the 3-4 days preceding the events. However, for HPE in southern France, Duffourg and Ducrocq (2013) estimated that evaporation from the Mediterranean represents only about 40% of the 175 water vapor feeding deep convection. Remote sources may also supply moisture, especially for the most intense HPEs, such as the Atlantic (Pinto et al., 2013; Winschall et al., 2012) or tropical areas (Turato et al., 2004; Chazette et al., 2016), even in the form of atmospheric rivers (Davolio et al., 2020).

Overall, the Mediterranean Sea is a significant heat and moisture source (Duffourg and Ducrocq, 2011) and air–sea exchanges play a key role during HPEs (Lebeaupin Brossier et al., 2008; Rainaud et al., 2016). These exchanges are 180 particularly intense in autumn; they are expressed in terms of turbulent fluxes of heat, moisture and momentum, and are controlled by gradients of temperature, humidity and wind velocity, respectively, at the air–sea interface. The air-sea interactions can modify the low-level atmospheric stability, and notably impact the flow regime and the location and intensity of convergence features and of atmospheric convection and precipitation (e.g. Homar et al., 2003; Xie et al., 2005; Rainaud et al., 2017); thus, SST values and patterns can influence the structure and organization of precipitation systems, 185 their lifecycle, severity, propagation speed, and track (Cassola et al., 2016; Bouin et al., 2017; Stocchi and Davolio, 2017). Additionally, ocean waves may significantly impact roughness length, thus momentum fluxes and low-level dynamics of the atmosphere, influencing the atmospheric planetary boundary layer (PBL) and the localization of HPEs (Thevenot et al., 2016).

The mechanisms of HPEs for the three target areas are described below, highlighting the contribution of the HyMeX IOPs to 190 their better understanding.



## 3 HPEs over the Italian target areas: outcomes from HyMeX

### 3.1 North-eastern Italy

The presence of a pronounced trough or a cyclone over the WMED, approaching the Italian peninsula from the west, is the
typical atmospheric configuration responsible for most HPEs over the north-eastern Italian area during autumn; this
configuration is associated with intense low-level southeasterly flow over the Adriatic Sea (Sirocco), which conveys
moisture towards the Alps (Manzato et al., 2015). IOPs 18 and 19 were both characterized by this large-scale pattern and,
together with IOPs 2 and 13, also highlighted another interesting dynamical local process in the area. In fact, when the
mesoscale setup described above is established, an easterly-northeasterly flow appears over the eastern Po valley, just ahead
of the Alpine reliefs, resulting from the deflection of the low-level jet by the orography. This barrier wind, which often
anticipates severe weather events, was already observed (radiosoundings) and simulated, and is a consequence of the low-
level blocking of the incoming southeasterly flow from the Adriatic, enhanced by the presence of cold air over the plain
resulting from nocturnal radiative cooling (Davolio et al., 2009; Buzzi et al., 2020b). However, the SOP1 campaign allowed
to better and more clearly identify its important role in triggering heavy precipitation where the barrier wind converges with
the Sirocco. This behavior critically depends upon the thermodynamic characteristics of the impinging southerly flow that
determines the evolution of the convergence pattern, as described in details in the following.

On the one hand, the HPEs affecting the north-eastern Alps - "Alpine" HPEs described and analyzed in Davolio et al. (2016)
and Stocchi and Davolio (2017) - were provoked by the transition from a "flow around" to a "flow over" regime following
the intensification of the Sirocco, which progressively penetrates inland, removes the barrier wind (and the convergence) and
rises over the orography (Fig. 3). This evolution is generally favored by a nearly moist neutral profile in the low levels, and
is characterized by low values of the Froude number (Rotunno and Ferretti, 2001; Miglietta and Rotunno, 2005, 2006).
Convection is inhibited upstream of the Alps, but once the moisture-laden flow is lifted over the mountains, convection can
be triggered, embedded within the stratiform orographic precipitation and locally increasing its overall intensity. Long-
lasting events of this type are often characterized by extreme rainfall amounts, as was the IOP19 and the case of October
2018 infamous Vaia storm (Cavaleri et al., 2019; Davolio et al., 2020; Giovannini et al., 2021).

On the other hand, when the incoming marine flow is conditionally unstable, the blocked flow situation persists, resulting in
low-level convergence well upstream of the orography (Fig. 3) that may evolve in the development of deep convection
("Upstream" HPEs in Davolio et al., 2016) or even mesoscale convective systems (Davolio et al., 2009; Ricchi et al., 2021)
and supercells (Manzato et al., 2015; Miglietta et al., 2016), which produce heavy rainfall and hail over the plain, even close
to the coastal areas (Manzato et al., 2020). In this category (e.g., IOP18), characterized by high values of the Froude number,
other parameters defined by the vertical profile provide an indication of the possible evolution and severity of the event. In
agreement with idealized numerical experiments (Miglietta and Rotunno, 2009, 2010), convection is triggered when the level
of free convection (LFC) is located at a low altitude, so that the uplift of the Sirocco over the barrier wind, in correspondence
with the convergence, is sufficient to overcome it. Consequently, Sirocco does not penetrate inland, but begins to feed the

convective precipitation systems. Therefore, the ratio between the depth of the cold air layer and the LFC is indicative of the possible triggering of convection over the plain (Davolio et al., 2016).

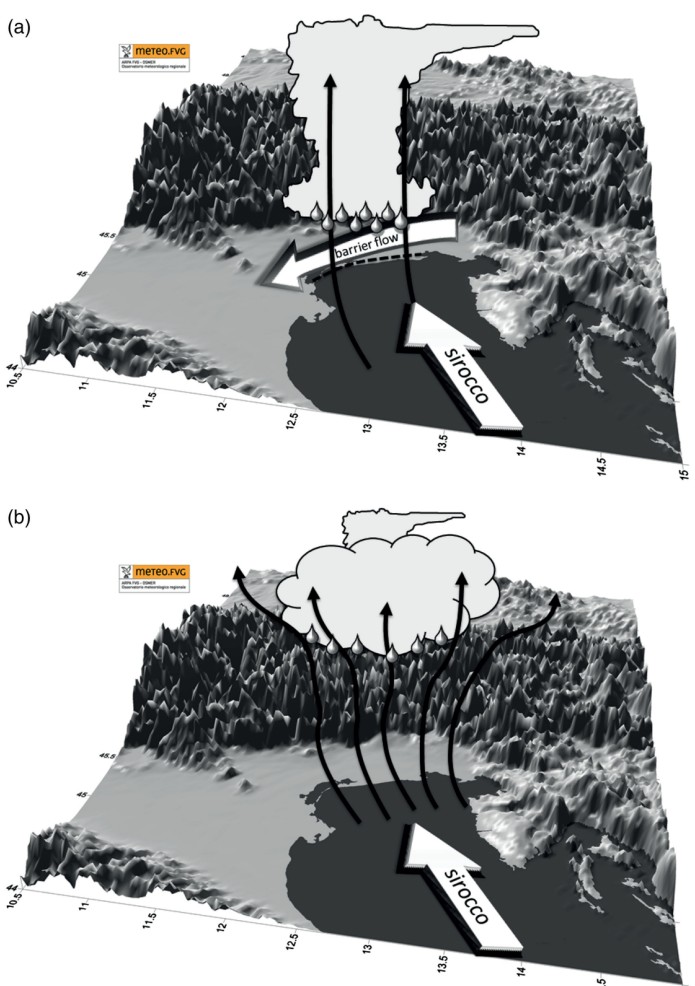

Figure 3: schematic diagram of the key mechanisms responsible for the two different precipitation patterns over NEI. (a)
Upstream event: blocked low-level flow, barrier wind, convergence and deep convection development occurring over the plain, upstream of the orography. (b) Alpine event: unblocked low-level flow, flow-over conditions, orographic lifting and precipitation over the Alps with possible embedded convection (From Davolio et al., 2016).

The first type of events received a lot of attention and was widely investigated in the last decades in several projects and field
campaigns (MAP, Bougeault et al. (2001) among others). Being mainly characterized by strong synoptic and orographic forcing, these events benefit of a relatively higher predictability (Grazzini et al., 2007; Cavaleri et al., 2019). Conversely, when convection takes place, the interaction of small-scale convective and turbulent processes strongly limits the model



prediction capability, since intrinsic predictability hardly exceeds the timescale of few hours or even less. However, the mesoscale organization of the flow, as in the presence of a convergence line that triggers convection, may account for a

somewhat extended range of predictability with respect to spontaneous isolated convection (Buzzi et al., 2014).

Therefore, most of the research efforts in HyMeX was devoted to attain a better understanding of convective episodes in order to finally improve forecast accuracy, or at least to increase awareness of the processes that anticipate the possible development of severe weather systems. In this framework, the Friuli Venezia Giulia (FVG) region is a suitable natural laboratory, since it does not only display the highest peak in the average yearly rainfall in the Alpine region (Isotta et al.,

2014), but also a high frequency of severe convective episodes as thunderstorms (Feudale and Manzato, 2014; Pucillo et al., 2020), hailstorms (Manzato, 2012; Punge et al., 2014), tornadoes and waterspouts (Giaiotti et al., 2007), and bow-echoes (Pucillo and Manzato, 2010).

IOP2 allowed a detailed study of two intense convective systems evolving into a supercell (Ferretti et al., 2014; Manzato et al., 2015; Miglietta et al., 2016) over the northeastern plain. A synoptic trough determined diffluent flow in the middle

troposphere over NEI (a common worldwide condition favoring HPEs; see Pontrelli et al., 1999, Lin et al., 2001) and drove a very warm and moist southeasterly low-level jet (another ingredient favorable to HPEs) from the Adriatic. The presence of low equivalent potential temperature ($\theta$e) air aloft determined potential instability in the eastern Po Valley, with values of CAPE well above the average for the period. The characteristic convergence pattern previously described was enhanced by an additional relatively dry southwesterly wind over the Italian northern Adriatic coast, originated as downslope Apennines

foehn. Moreover, the presence of a shallow low-pressure system over the Venice lagoon maintained the high-$\theta$e low-level flow from the sea in the vicinity of the foothills, where convection was triggered, for many hours (Fig. 4). The correct simulation of the storm evolution turned out to be critically dependent on the intensity of this small-scale cyclone and on the low values (but greater than zero) of convective inhibition: both features were necessary to allow the release of convection only near the foothills (Miglietta et al., 2016). The evolution of the mesoscale convergence features controlled the

displacement of the supercell in a way qualitatively different from the classical supercell conceptual model developed for the US Great Plains, which are characterized by more homogeneous and stationary conditions (Rotunno and Klemp, 1985).

SOP1 activities allowed also to exploit the synergy among modeling simulations and monitoring facilities, showing the capability of detecting and simulating important features of convective storms, useful also in operational applications. For example, in IOP2 the Doppler radar identified a mesocyclonic circulation, suggestive of supercell development, in agreement

with the numerical simulations. Following a phase of complex interaction between two storms (Miglietta et al., 2016), one cell evolved into an arc-shaped echo in the VMI (Vertical Maximum Intensity) radar field (Fig. 5), in agreement with the expected theoretical evolution of an extensive downburst into a bow-echo-like structure (Fujita, 1978). Similar evolutions over the area, in particular over the Grado Lagoon, sustained by strong cold air rear inflow, have been recently reported in other episodes of severe convection in the region (Pucillo et al., 2020). Hence, the latter emerges as a common mechanism

for the intensification of deep convection, occurring when the warm and moist air near the Venice lagoon and the northern Adriatic Sea contrasts with cooler air behind a small cold frontal system moving eastward.





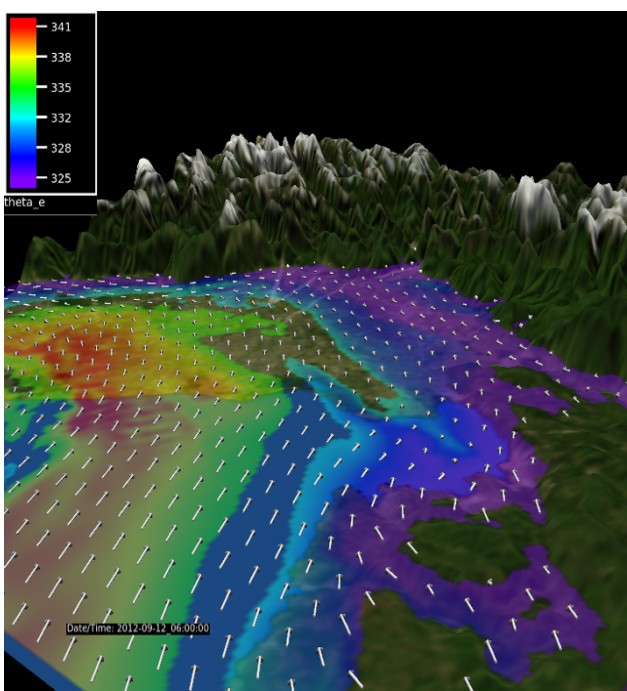

Figure 4: Wind vectors at 350 m height (white arrows), equivalent potential temperature at 300 m (shaded) at 0600 UTC, 12
September 2012 (IOP 2), from a run initialized with the GFS analysis at 1200 UTC, 11 September (From Miglietta et al.,
2016).

As described in Section 2, one of the main results of HyMeX concerns the identification of the evaporative cold pool under
an MCS as one of the key development mechanisms. This feature has been shown for many systems triggered over the sea or
close to the coast in other areas (Lee et al., 2016; Lee et al., 2018, Duffourg et al., 2016; Bouin et al., 2017), where the
interaction of the cold pool with the incoming moist flow is essential for the subsequent triggering of convection and
determines the evolution of the MCS. In contrast, over NEI, where high-θe low-level flow is channeled along the Adriatic
towards the Alps, the cold pool is able to modify the propagation and position of convection only slightly, but it does not
determine the stationarity of the systems, for which the primary mechanism is the persistent convergence of the low-level jet
with the cold barrier wind in blocked-flow conditions (Davolio et al., 2016). Only in the case of supercell development, the
cold pool on the forward flank determines the movement of the system, as well as the baroclinic generation of vorticity along
the cold-air boundary and its intrinsic low-level rotation (Rotunno and Klemp, 1985).

Finally, Miglietta et al. (2016) showed that the low-level cold air trapped in the narrow Alpine valleys can be crucial to limit
the northward extent of the warm air tongue. Due to their relatively coarse resolution, global analyses/forecasts often miss
this feature, possibly leading to the significant climatological underestimation of the rainfall simulated by ECMWF forecasts



in the FVG plain and coastal area during summer (Manzato et al., 2016). In turn, the presence of cold-air damming may easily be missed or misrepresented in the initial condition of mesoscale models, too.

Figure 5: Vertical Maximum Intensity (VMI) of the reflectivity measured by the Fossalon di Grado radar, with θe (red–blue color bar in dBZ) and 10 m wind observed by surface stations (5 min time resolution) and the CESI-Sirf cloud-to-ground lightnings within the ± 6 min interval (color bar shows time delay in minutes) at 0820, 0850 and 0950 UTC, 12 September 2012; bottom left: Fossalon radar maximum reflectivity at 0840 UTC vertical (VMI) and lateral projections (HVMI) (Reprinted from Atmospheric Research, 153, Manzato et al. 2015. 12 September 2012: A supercell outbreak in NE Italy?, 98–118, Copyright (2015), with permission from Elsevier).

## 3.2 Liguria and Tuscany

A gridded climatology for Italy has recently confirmed that LT is one of the wettest regions of the peninsula, second only to the north-eastern Alpine area (see Fig. 12 in Crespi et al., 2018). In particular, precipitation affects mainly the Liguria region and the northern part of Tuscany, characterized by the highest and steepest orography. The region is prone to severe floods,





as demonstrated by the large number of HPEs in the recent past that produced significant damages (Buzzi et al., 2014;
Davolio et al., 2015b; Cassola et al., 2016; Silvestro et al., 2016; Fiori et al., 2017 among others). This is due to its peculiar topographical characteristics (steep concave-shaped orography that reaches high elevations within a few kilometers from the coast) that make it exposed to southerly moist flow, impinging the region in the form of low-level jets, sometimes as atmospheric rivers (Davolio et al., 2020) or warm conveyor belts (Bertò et al., 2004). Without claiming to cover all the possible types of HPEs affecting LT area, HyMeX outcomes and subsequent studies identified and focused on two separate
classes of events:

- intense orographically-enhanced precipitation over the Apennines;
- MCSs formation over the sea or close to the coast, that include a large portion of severe weather episodes and represent a serious threat for the region.

Each class presents some typical recurring features and both share most of the large-to-mesoscale ingredients necessary to
generate persistent rainfall that may evolve into a devastating flood in a few hours.

The occurrence of severe weather episodes in the area is due to a combination of processes at different scales. On the synoptic scale, a slowly advancing baroclinic disturbance associated with an upper-level diffluent trough over NW Med and an upper PV maximum are known as necessary (but not sufficient) ingredients for heavy precipitation (Nuissier et al., 2011). Sometimes, orographic cyclogenesis in the lee of the Alps (Buzzi et al., 2020a) plays a role too. In common with other
coastal areas of WMED, the slowly evolving large-scale circulation favors, at the mesoscale, the persistence of frontal features and of a low-level moist and conditionally unstable airflow directed towards the coastal mountains, whose vertical humidity profile is critical in determining the persistence of torrential rain in a specific area (Duffourg et al., 2018; Lee et al., 2018). However, the most recent research focused on meso-$\beta$ and meso-$\gamma$ scale processes (typical of density current dynamics and convective-scale auto-organization), whose prediction and knowledge are affected by greater uncertainties.
Quasi-stationary V-shaped MCSs have been the main topic of investigation, considering that they were responsible for some of the worst floods in the last century (Silvestro et al., 2012; 2016). High-resolution modelling studies, supplemented by available observations, have identified the peculiar dynamical process responsible for the triggering of convection, i.e. the convergence of two low-level currents over the Gulf of Genoa (Buzzi et al., 2014; Fiori et al., 2014). On the one hand, a high CAPE, moisture-laden southerly low-level jet channels between Corsica and Tuscany coast; on the other hand, a low-level
flow of cold and dense air spills through the lowest gaps of the Apennines in the central part of Liguria and propagates southward as a density current over the Gulf of Genoa (Fig. 6). The latter flow originates from pre-existing cold air in the western Po Valley, drawn towards the sea by an ongoing pressure drop; once it reaches the coastline, its thickness does not exceed 1 km, becoming progressively thinner as it moves offshore (Fiori et al., 2017). The convergence produces an uplift of the low-level moist air, sufficient to overcome the LFC and thus to allow the release of convective energy (Fiori et al., 2014;
Davolio et al., 2015b). It is worth noting that during the MAP field campaign in 1999, this kinematic interaction was suggested and somehow observed through airborne and ground-based radar observations (Bousquet and Smull, 2003), but modeling and monitoring tools available at the time were not yet suitable for the recognition of the small-scale features



described above. Once the MCS has formed and convection has fully developed, the downdrafts can interact and reinforce the pre-existing low-level cold flow. The equilibrium between the intensity of the overall cold flow and the southerly jet, and
the instability of the air transported by the jet determines the exact evolution of the precipitation system and the location and duration of the rainfall event.

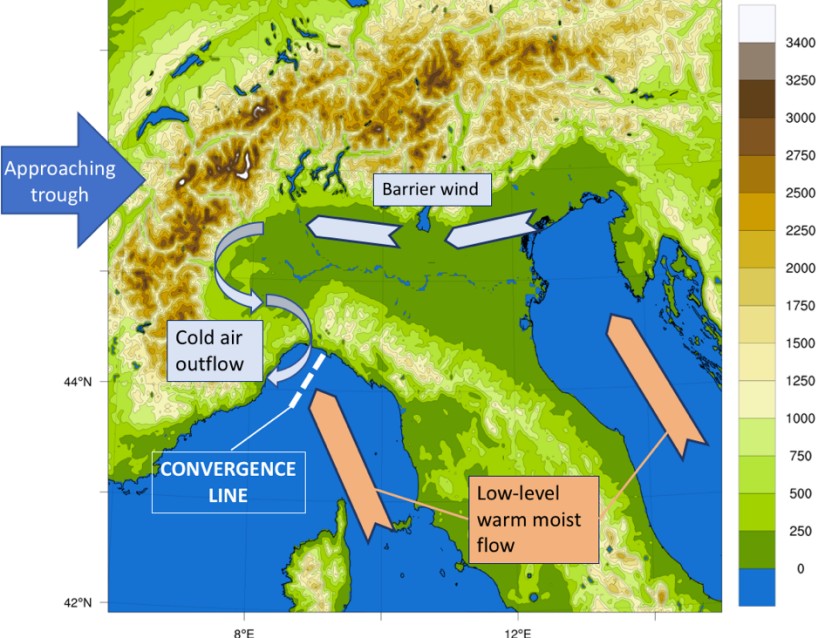

Figure 6: Sketch of the mechanisms for the triggering of quasi-stationary MCS over the Gulf of Genoa.

Of course, the correct quantitative precipitation forecast (QPF) strongly depends on the model's ability to represent the cold air outflow from the Po Valley and the convergence setup (Buzzi et al., 2014). Consequently, the NWP spatial resolution becomes an important factor (Davolio et al., 2015b; Fiori et al., 2017), because it allows to describe with precision not only the convective system dynamics, but also the orographic features (Bassi, 2014), and in turn the propagation of the density current responsible for the convergence line. Buzzi et al. (2014) concluded that the crucial role played by this mesoscale
setting can explain the extended range of predictability of these events with respect to the typical timescale of isolated and spontaneous convection. Although this convergence pattern does not assure that heavy precipitation occurs, it has certainly become a warning sign for weather forecasters.

A well-observed example of this kind of events was monitored during IOP16a. One of the several MCSs developed in the northern part of WMed (Duffourg et al., 2016) was triggered over the eastern Ligurian coast, as a consequence of low-level
convergence. The pre-frontal MCS remained quasi-stationary during the morning, producing precipitation amounts reaching 250 mm/24h and up to 50 mm/1h, that were responsible for local floods. Nuisser et al. (2016) confirmed the important role

played by the low-level convergence using two convection-permitting ensemble systems, showing that it is a key factor controlling the predictability of HPEs over the Liguria region.

An additional important first-time confirmation of this dynamical mechanism was provided by the dropsonde data during the last IOP of the SOP1. In fact, during IOP19, two dropsondes were deployed during the Falcon flight over the Gulf of Genoa, respectively over the western and the eastern part (Fig. 7). While the latter recorded southerly to southwesterly flow all along the profile, the former revealed a sharp low-level thermal inversion, below 900 hPa, associated with the presence of a northerly, stably stratified flow advecting cooler air from the Po Valley.

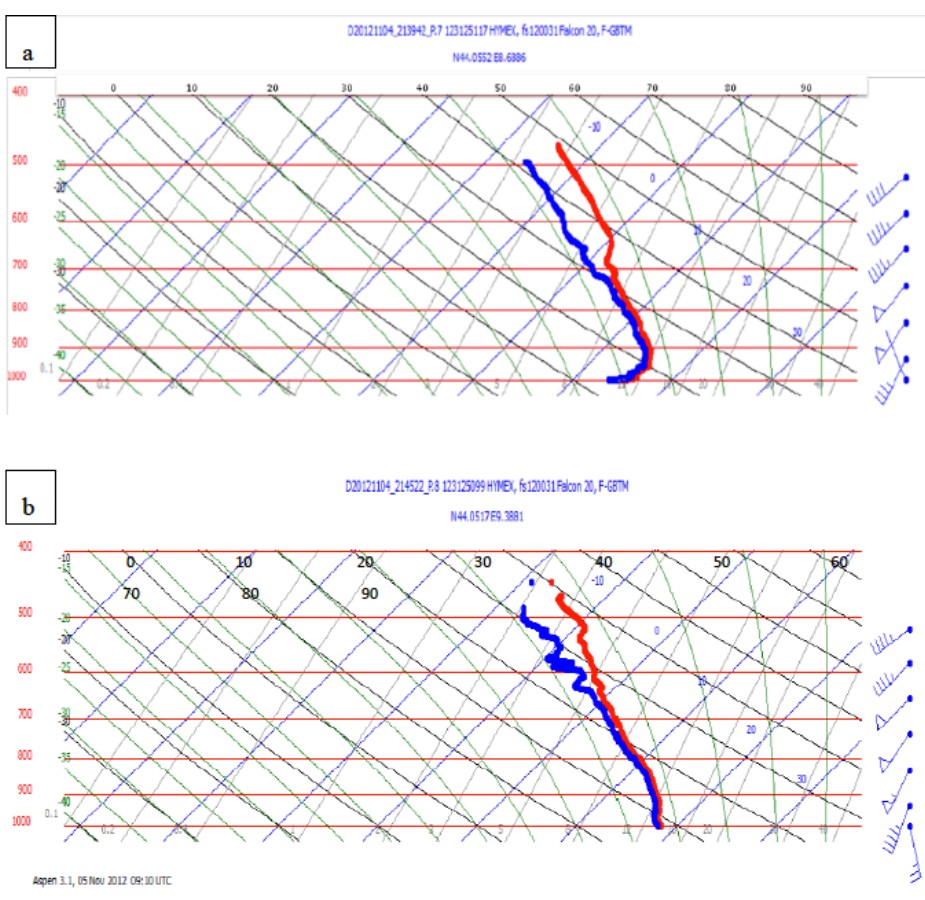

Figure 7: Vertical profile from the two dropsondes deployed by the Falcon flight over the Gulf of Genoa at (a) DS1 (44.055° N, 8.689° E) at 2140 UTC, and (b) DS2 (44.052° N, 9.388° E) at 2145 UTC, 4 November 2012. Courtesy of Julien Delanoe (LATMOS/IPSL/UVSQ, France) (From Ferretti et al., 2014).

However, during IOP19 convection did not develop over the sea, in spite of the presence of low-level convergence. Instead, intense orographic precipitation occurred over the Apennines between Liguria and Tuscany, fed by intense southwesterly flow induced by a cyclone over the Gulf of Lyon. Therefore, in this case the role of the orography was two-fold. On the one





hand, it modulated the gap flow directed southward over the sea, which restrained the southerly inflow on the eastern portion of the gulf of Genoa, and produced local convergence; on the other hand, it produced the uplift of the humid impinging flow. Both forcings were critical in determining the location and intensity of heavy rainfall and were generally well reproduced by

high-resolution models (Ferretti et al., 2014). Orographic precipitation occurred also during IOP7b and IOP16c, but associated with embedded convection (as revealed by lightning data) that enhanced the rainfall intensity in correspondence with the orographic divide. These mechanisms were already thoroughly studied in the past (e.g., Rotunno and Houze, 2007); however, in this context, it was possible to show that high-resolution, convection permitting models provide a useful guidance for forecasters, and that their performance clearly improves as grid-spacing decreases (Bassi, 2014; Ferretti et al.,

2014; Davolio et al., 2015b; Barthlott and Davolio, 2016).

The relevant dynamical aspects that affect the heavy precipitation over LT are due not only to the orography of the region, which provides the direct uplift or blocking (Alps and Apennines), but also to the main upstream islands (Sardinia and Corsica). In IOP13, Barthlott and Davolio (2016) showed that the islands can perturb significantly the low-level wind, temperature and humidity fields. Thus, the flow deviations (due to blocking, flow splitting, and channeling effects between

the two islands) and local circulations (including drainage flows, land and sea breeze systems) determine either patterns of convergence over the sea upstream of the Italian coast, where convection is triggered, or the exact location where the deflected flow interacts with the orography inland. Moreover, the offshore night convection around Corsica modifies the conditions upstream of Tuscany and may affect precipitation there (Barthlott et al., 2016).

The feedbacks between convective-scale motions and orographic flows can however be very complex; hence, it is not

straightforward to trace back the observed HPEs to theoretical studies in idealized conditions, such as Miglietta and Rotunno (2014) or Kirshbaum et al. (2019). Case study analysis may highlight important physical aspects or point out predictability/forecasting issues. During the SOP1, the development of convection between the Apennines and the Po Valley was analyzed in Pichelli et al. (2017), comparing IOP6 and IOP13. In both cases an eastward moving trough (and front) induced organized convection in the region; however, the slightly different characteristics of the meridional winds across and

ahead of the front determined a delicate balance between the competing mechanisms of orographic subsidence on the lee side of the Apennines and frontal uplift in the Po valley, responsible for the different degree of predictability of the two cases. In particular, for IOP13, the "flow around" regime contributed to low-level convergence in the Alpine concavity; for IOP6, the flow upstream was at the threshold between "flow-over" and "flow-around" regimes, hence downstream convection was very sensitive to small differences in the simulated environmental parameters.

Rainaud et al. (2016) found that, in the whole SOP1, although some HPEs occurred without significant air–sea fluxes, all strong air–sea exchange events occurred during or 1-2 days before an HPE. IOP13 allowed to show how air-sea exchanges, even in areas relatively far in the Mediterranean or even few days in advance, may play a key role for HPEs over LT. A persistent southwesterly flow within the marine boundary layer transported moisture towards the French and Italian coast from the western Mediterranean, feeding convective systems. The same moisture was extracted by the dry and strong Mistral

over the Gulf of Lyon 1-2 days before and advected near the periphery of the Mediterranean (see Figs. 13 and 14 in Rainaud





et al., 2016). This means that there can be a connection between apparently independent severe weather events, manifesting through air-sea interaction processes. Caldas-Alvarez and Khodayar (2020), assimilating GPS-derived Zenith Total Delays (GPS-ZTD) data in sensitivity experiments for IOP16a, showed that the residual moisture of a previous HPE may be transported by a large-scale front and contribute, together with evaporation from the Mediterranean and north Africa, to

precipitation.

### 3.3 Central Italy

The CI hydro-meteorological site is located in a peculiar position, being surrounded by the sea on both sides, the Tyrrhenian Sea to the west, the Adriatic Sea to the east. This area is densely populated especially because it includes the wide urban area

of Rome, with approximately 4 million inhabitants. As the rest of the peninsula, CI is characterized by complex orography: moving from the western to the eastern coast across the region, the elevation grows from sea level to nearly 3000 m in less than 150 km and then down again. This area is characterized by many small- to medium-sized steep and densely urbanized coastal catchments (Lombardi et al., 2021), that display fast response to heavy precipitation and are more prone to flash floods.

Two main synoptic patterns may be identified as favorable to heavy rainfall over CI, with precipitation systems affecting either the western or the eastern side, fed by moist low-level currents flowing over the Tyrrhenian or the Adriatic Sea, respectively:

- in the first category, an upper-level trough (and possibly an orographic cyclone over the Gulf of Genoa) and its associated cold front approach the Italian peninsula from the west; convective systems are triggered over the sea and therefore affect CI,

as in the case of IOPs 12 and 13;

- in the second category, the presence of a cut-off low over the Tyrrhenian Sea is responsible for a cyclonic circulation over the central Mediterranean, which induces an intense Bora flow over the Adriatic Sea, whose interaction with the Apennines orography produces intense rainfall, as occurred during IOP4.

Both situations were studied in details during HyMeX through numerical modeling activities and many instruments deployed

over CI, dedicated in particular to the study of the microphysical properties of convective precipitation. In fact, besides the operational network, which comprises more than 200 raingauges and three C-band radars, also X- and K-band radars, and different types of disdrometers were specifically installed for the campaign (details in Ferretti et al., 2014).

IOP13 was characterized by a quite strong synoptic forcing, with an upper-level trough extending toward northern Algeria and moving eastwards towards Corsica and the Tyrrhenian coast. Ahead of the trough, the cold front shifted over the

Mediterranean Sea, moving around a low pressure centered over the Gulf of Genoa. A pre-frontal intense warm and moist ($\theta_e$ above 330K) airflow from southwest between Sicily and Sardinia impinged upon the Tyrrhenian coast; its convergence with westerly winds, blowing at higher latitudes, allowed the triggering and development of an intense convective line that affected the urban area of Rome. In these conditions, numerical models reproduced quite accurately the convective system,



which grew within the warm sector of a Mediterranean cyclone, and the associated intense rainfall. However, the correct
forecasting of the precipitation systems was dependent on the accurate description of the fine structure of the cyclone
(Ferretti et al., 2014).

On the same day, a multi-cell, V-shaped, retrograde-regeneration MCS developed over the Tyrrhenian Sea, but affected an
area slightly to the south of CI. Lee et al. (2016) showed that, besides the low-level convergence over the sea, ahead of the
upper-level trough (see their Fig. 11a), and the low-level advection of warm and moist air from south, the strength of
convection was increased by an elevated moisture plume from the Tropics that enriched the mid-level humidity content (Fig.
8). Later, a second MCS was triggered by the orography of Algeria and affected Sicily and southern Italy, again maintained
by the high low-level moisture content and a marked convergence. In order to improve the simulation skill for the latter
MCS, Lee et al. (2016) found that the inclusion of northern Africa in the domain was crucial.

Barthlott and Davolio (2016) concluded that predictability of MCSs is case dependent: lower for isolated convection systems,
as the former, higher when they are directly triggered by the frontal passage, as the latter. They thoroughly analyzed the
different phases of IOP13, finding that the low-level flow is crucial for HPEs over CI, while its thermodynamic properties
are essential to determine the interaction with the orography. In fact, intense precipitation affects the internal area only if the
flow is able to overcome the Apennine reliefs, while a stable impinging flow is deflected to the north along the coast of CI.
Consequently, models succeed in forecasting heavy precipitation as long as they correctly describe the low-level flow
characteristics.

Not only the low-level moisture advection towards coastal areas is important, but also the total atmospheric moisture was
identified as a key parameter for convective initiation (Khodayar et al., 2018). The relevance of moisture profiles for HPEs
emerged in several IOPs and for different WMED target areas. For instance, during IOP12, heavy precipitation systems
developed within an area with maximum integrated water vapor (IWV) of 35–45 kg/m$^2$; also, the most intense events
received moisture from different sources simultaneously (i.e., WMED and Northern Africa), thus involving transport on
different time-scales. A sudden increase in IWV of about 10 kg/m$^2$ occurred 6-12 h before the intense events; in particular,
the increase in the low-level humidity content strongly impacted on stability and convection intensity (Khodayar et al., 2018).
The synergy between the different observation platforms and the lightning detection network allowed, for the first time in
Italy, to simultaneously investigate several aspects of convection such as lightning (LINET network, Betz et al., 2009;
Federico et al., 2014; Torcasio et al., 2021), raindrop size distributions at the ground (disdrometers), and graupel aloft (radar).
Roberto et al. (2016) confirmed the correlation between the mass of graupel and lightning activity (number of strokes), an
important aspect that should be considered in order to improve the modeling of convection. Moreover, instruments deployed
during the SOP1 contributed to a long time series of disdrometer measurements in Rome, that allowed to optimize radar
algorithms specifically for the Italian climatology, and to improve the estimation of vertical profiles of rain drop size
distribution (Adirosi et al., 2016; 2018).



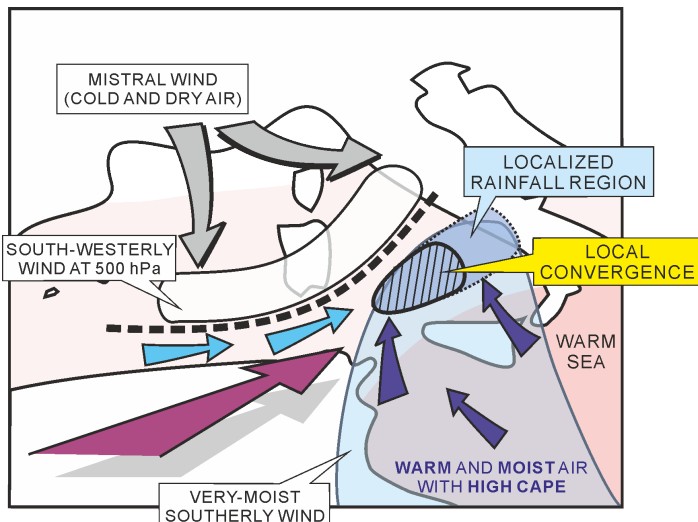

Figure 8: Schematics summarizing the main features and processes responsible for the maintenance of deep convection upstream of Southern Italy and leading to HPEs for Period 1 of HyMeX IOP13. Grey arrows indicate the mistral wind behind the edge of the cold front (thick dashed line). The light blue arrows depict the low-level westerlies-to-southwesterlies ahead of the front. Dark blue arrows show the low-level moist southerlies from the eastern Mediterranean encompassed in the blue-shaded area. The purple arrow illustrates the elevated tropical plume. The local low-level convergence is indicated by a hatched area. The rainfall regions are indicated by blue shading within the dotted line (From Lee et al., 2016).

Despite being a relatively small and shallow basin, the Adriatic Sea plays a critical role during HPEs affecting the CI. This occurs mainly through air-sea exchanges (Ricchi et al., 2016; Licer et al., 2016; Davolio et al., 2017), which may be very intense during Bora events. The presence of a low-pressure system settled over the central Mediterranean induces cyclonic Bora (Horvath et al., 2009) over the northern Adriatic basin; the latter is often accompanied by precipitation and gusty winds affecting the eastern side of the CI target area. This was the case of IOP4, occurred on 14 September 2012, when heavy 485 precipitation (more than 150mm in a few hours), mainly along the central Italian coast, caused several river overflows and flooding (Mazzarella et al., 2017). Convective systems in the central Adriatic basin (Fig. 9) were triggered by a low-level wind convergence between northeasterly cooler Bora and moist and warm southeasterly Sirocco, and were later advected inland. Furthermore, orographic uplift over the eastern slopes of the Apennines produced stratiform precipitation (Fig. 10). Balloons and radiosounding observations in L'Aquila indicated a high humidity content in the lowest 6 km in IOP4, along 490 with a lowering of the tropopause, as shown by the $O_3$ measurements. Such conditions are expected in the case of an intense Mediterranean cyclone close to the area and of intense Bora conditions. In addition to the simplistic conceptual model that associates rainfall to the mesoscale circulation around a low-pressure system and to the orographic uplift of moisture, Davolio et al. (2017) showed that the interplay between surface fluxes and orographic effects is complex and basically





nonlinear. In fact, heat and moisture exchanges at the Adriatic Sea surface impact the PBL thermodynamic profile and

stability, which control the dynamical processes induced by the downstream orography:

-    in case of "flow over", upstream ascending motion produces heavy precipitation along the Apennines slopes and
     over the crest;

-    in case of blocked-flow regime, flow retardation and deviation are responsible for precipitation localized upstream
     of the ridge divide (Stocchi and Davolio, 2017).


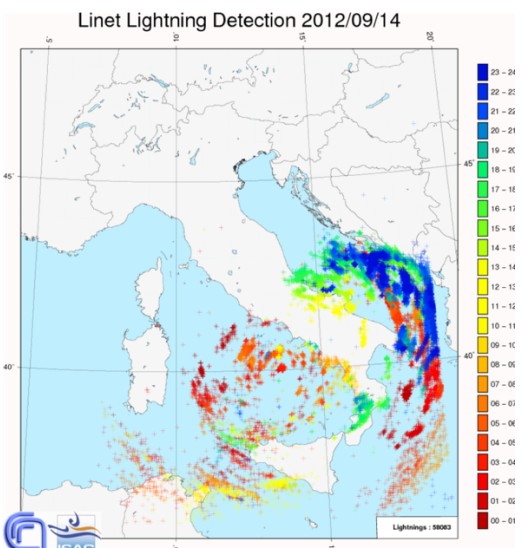

Figure 9: LINET lightning activity measured on 14 September 2012. The map shows the intra-cloud and cloud-to-ground
strikes registered in 24 h. Different colors are associated with different hours.

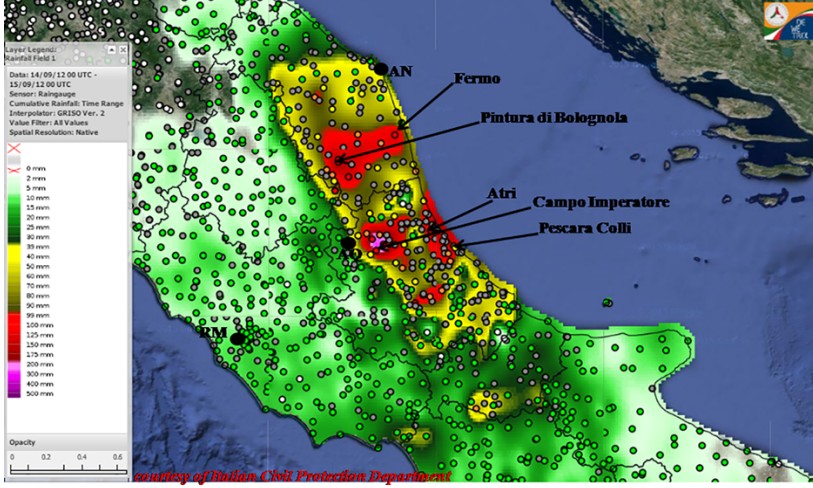






Figure 10: Interpolated map of 24 h accumulated rainfall from 0000 UTC on 14 September 2012 over Abruzzo and Marche regions taken from the DEWETRA system from rain gauge measurements. Black contours are the administrative boundaries of the regions, while the colored circles represent the warning pluviometer thresholds (From Maiello et al., 2017; Courtesy of Italian Department of Civil Protection).

## 4 Summary and Conclusions

The first Special Observation Period of HyMeX was held in fall 2012 and focused on the observation and numerical simulation of HPEs and floods in the northwestern Mediterranean. Nine of the twenty IOPs involved Italy, enabling a unique opportunity for monitoring and analyzing HPEs over the three Italian target areas of NEI, LT, and CI. The present paper provides a review of the main findings emerged from the HyMeX campaign over Italy, allowing for an improved understanding of the mechanisms responsible for HPEs and for the identification of the most relevant mesoscale features.

In NEI, the SOP1 campaign allowed to clarify the important role of the barrier wind, resulting from the blocking of the southeasterly moist and warm low-level flow from the Adriatic, in triggering heavy precipitation. Two main patterns responsible for HPEs were identified, depending on the upstream environmental conditions, in particular static stability and wind speed:

- Upstream events, characterized by low-level blocked flow, barrier wind, convergence and deep convection over the plain, even well upstream of the orography;

- Alpine events, in which the intensification of the southerly low-level jet progressively erodes the barrier wind, and produces an unblocked low-level flow, orographic lifting and precipitation over the Alps, possibly associated with embedded convection.

The analysis of IOP6 revealed also that the numerical simulation of convection in the Po Valley is particularly sensitive to small changes in the upstream environmental conditions when they are close to the threshold between "flow-over" and "flow-around" regimes (Froude number ~ 1).

Some typical ingredients conducive to severe convection in the region were identified in IOP2. In particular, a tongue of moist air advected toward the eastern Po Valley and curved westward by a shallow pressure low in the northern Adriatic was found to play a key role in the development of convective storms near the foothills and in their subsequent evolution into small bow-echo-like structures in the coastal areas.

LT is one of the wettest areas of the peninsula and in the last decade it has been severely affected by a high number of HPEs and disastrous floods, favored by the steep concave-shaped coastal orography exposed to southerly moist low-level jets. HyMeX focused on two separate classes of events:

- intense orographically-enhanced precipitation over the Apennines;

- quasi-stationary MCSs over the sea or close to the coast.



The latter category, which includes a large part of HPEs in Liguria, was deeply investigated in HyMeX (e.g., IOP16a). Convection is typically triggered by the convergence of two main low-level currents: a persistent moist and conditionally unstable flow channeled between Corsica and Tuscany coast over the Tyrrhenian Sea; the outflow of cold and dense air,

spilling from the Po Valley through the lowest gaps of the Apennines, forced by an ongoing pressure drop, and propagating southward as a density current over the Gulf of Genoa. The latter was clearly identified as a sharp low-level inversion in a dropsonde profile over the western side of the Gulf of Genoa during IOP19. The equilibrium between the southerly jet and the intensity of the cold flow, possibly reinforced by convective downdrafts, as well as the instability of the warm air impinging on the cold pool determine the exact evolution of the MCS.

The HPEs over LT are determined not only by the orography of the region, but also by the perturbations induced by Sardinia and Corsica on the environmental parameters. Dynamic forcing and thermally-driven circulations may deflect the flow, modifying the areas where it interacts with the orography, or influencing the convergence patterns upstream of the Italian coasts where convection is triggered.

Finally, two main synoptic configurations were identified for HPEs in CI target area:

- when an upper-level trough approaches the Italian peninsula from the west, low-level convergence over the sea triggers convective systems that later affect the Tyrrhenian coast, sustained by high low-level moisture content and, possibly, by a mid-level moisture plume from the Tropics, as in IOP13;

- when a cut-off low develops over the Tyrrhenian Sea, intense Bora blows over the Adriatic Sea and the eastern side of CI is affected by intense precipitation and gusty winds, as in IOP4. Convective systems develop over the sea, triggered by low-

level convergence, while orographic uplift over the eastern slopes of the Apennines produces stratiform/orographic precipitation.

The Adriatic Sea plays a critical role in the latter group of HPEs mainly through air-sea exchanges. The interplay between surface fluxes and orographic effects is nonlinear: sea surface exchanges may impact significantly the PBL thermodynamic profile and the stability of the flow impinging on the orography, favoring either "flow over" and precipitation over the

Apennines, or "flow around" and precipitation localized upstream of the ridge.

Many modeling studies agree that the moisture structure in the lower troposphere is a key factor for an accurate prediction of precipitation systems in the coastal mountainous region of the northern Mediterranean. In fact, the stratification of the lower troposphere, affected also by air-sea exchanges, determines both the interaction of the flow with the coastal orography and the location where convection is triggered.

In conclusion, the HyMeX SOP1 has allowed to disentangle some important mesoscale mechanisms conducive to HPEs in the central and northern Italy regions. In the near future, the TEAMx (Serafin et al., 2018) program will further clarify the mechanisms for orographic convection in the Alpine area with a dedicated Working Group. The planned activities will further improve our understanding of the processes relevant for HPEs in an area that has been the subject of intense investigation in the last twenty years (MAP; Bougealt et al., 2001), but shifting to focus on the meso-$\gamma$ scale processes.



In contrast, HPEs in southern Italy remain the subject of marginal attention in research activities and scientific literature, although intense precipitation systems are not unusual in the area (Caccamo et al., 2017; Martinotti et al., 2017). Funatsu et al. (2009) found that the arc between Tunisia, Sicily and southern Italy shows a climatological peak of convective activity in autumn, and is a prominent area of initiation of MCSs leading to HPEs over southern Italy. Moreover, due to its peculiar hydrography, i.e. short watersheds which are activated only in rainy periods and display fast response to precipitation, HPEs

may be conducive to flash floods (Senatore et al., 2020; Avolio et al., 2019) even if precipitation may be confined in very limited areas (e.g., Gascon et al., 2016). Quasi-stationary low-level convergence patterns (Mastrangelo et al., 2011), deep cyclones generating intense low-level wind associated with orographic uplift or convergence (Moscatello et al., 2008; Marra et al., 2018), orographic triggering over low obstacles in the presence of high low-level humidity (Miglietta and Regano, 2008) have been already identified as mechanisms potentially conducive to HPEs in southern Italy. Even in the absence of

dedicated funding, we hope that the Italian scientific community will be able to devote more attention to this area in the future, supporting the development of regional weather centers (belonging to the Department of Civil Protection) that have been established in southern Italy only recently.

**Code and data availability**
No data sets were used in this article.

**Author contributions**
MMM and SD jointly developed the review paper and edited the manuscript.

**Competing interests**
The authors declare that they have no conflict of interest.

**Acknowledgements**
This work is a contribution to the HyMeX programme. The authors are grateful to the Department of Civil Protection (DPC) for allowing access to the regional networks of raingauges, composite radar images and the DEWETRA visualization platform.

**Financial support**
This research has been supported by the project "FOE2019: Cambiamento climatico: mitigazione del rischio per uno sviluppo sostenibile", funded by the Italian Ministry of University and Research, and the by Project "DPC 2020-2021-Accordo DPC/CNR–ISAC"

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
