# Peer review of "Dynamical forcings in heavy precipitation events over Italy: lessons from the HyMeX SOP1 campaign"

_Hydrology and Earth System Sciences, 2021_

## Author Response (AR1)

*Comment on hess-2021-206*
*Anonymous Referee #1*
*Referee comment on "Dynamical forcings in heavy precipitation events over Italy: lessons from the HyMeX SOP1 campaign" by Mario Marcello Miglietta and Silvio Davolio, Hydrol. Earth Syst. Sci. Discuss., https://doi.org/10.5194/hess-2021-206-RC1, 2021*

*Major Comments*

*The manuscript presents a review of the lessons learned from the first special observation period of the HyMeX programme regarding heavy precipitation events (HPEs) in Italy. The main findings of this measurement campaign are contextualised and broken down separately for the three Italian target areas that were defined within the research programme. The text is clear and well-constructed; the references are complete and appropriate. The main conclusions are adequately set out.*

*In all, this is a very interesting article that provides an excellent synthesis of the state-ofthe-art knowledge of HPEs occurring in Italy and more broadly (but in less detail) around the western Mediterranean basin. I have only essentially technical remarks (see below) and I therefore recommend that the manuscript be accepted once these technical corrections have been made.*

We would like to thank the reviewer for the very positive evaluation of the manuscript and the constructive comments raised.

*Minor Comments*

*Although not much troublesome, the abstract is a bit long and the authors could consider shortening it.*

The Abstract has been shortened.

*Figure 2 could benefit from the addition of geographical references mentioned in the text that are not included in Figure 1 either, such as Algeria, Sardinia, etc.*

We added some additional geographical references in Fig. 1, including some regions in and around Italy that are mentioned in the text.

*There are some typos in the citations ("Nuissier", not 'Nuisser'; "Bougeault", not "Bougealt") as well as in the rest of the text ("these effort"; "Cévennes-Vivarais", not "Cévenne-Vivarais", "strokes", not "strikes"; etc.).*

Changed.

*Page 18, Lines 466–467: Can the authors clarify a little how the correlation between graupel and lightning may improve the modelling of convection? As currently written, the sentence is not very explicit.*

We agree that the sentence is ambiguous. Since this part is not strictly related to the dynamical forcing, which is the focus of the paper, we decided to remove it.

*Comment on hess-2021-206 (Reviewer)*
*Anonymous Referee #2*
*Referee comment on "Dynamical forcings in heavy precipitation events over Italy: lessons from the HyMeX SOP1 campaign" by Mario Marcello Miglietta and Silvio Davolio, Hydrol. Earth Syst. Sci. Discuss., https://doi.org/10.5194/hess-2021-206-RC2, 2021*
*The manuscript deals a challenging task of systematizing knowledge learned from 10 years of the Hymex experiment related research over Italian region, including dedicated SOP in 2012. The overview is given for three sub-regions in Italy which are known for heavy precipitation: northeastern Italy (NEI), Liguria-Tuscany (LT) and central Italy (CI) which were studied during the experiment.*
*The overview of dynamical forcings of HPE is sound and based on recent literature. However, there are a few relatively minor issues which need to be tackled before the manuscript can be deemed publishable:*

We would like to thank the reviewer for the positive evaluation of the manuscript and the constructive comments raised.

*14: Abstract specifies intensity of the impinging flow (i.e., the Froude number). Intensity of the flow and Froude number are related but are not directly associated, as Froude number (e.g. apart from orography) depends on the cross-mountain component of the flow. Please revise formulation or remove text in parenthesis.*

Text in parenthesis removed.

*108: Authors implicitly consider that Alps increase predictability and in theoretical context this is true. However, this is also model resolution dependent. Therefore I suggest to write "...thus, provided the sufficient numerical weather prediction resolution, they are adequately forecasted at least at short leading time.*

Changed following your suggestion.

*112-115: Low-level jet stream is also well known factor contributing to HPE. Consider stating explicitly here.*

Added.

*201: With this synoptic setup, due to flow splitting of northwesterly cold over the Alps, NE winds (bora) in the northern Adriatic are quite frequent. To include barrier wind into conceptual model, a clear demarcation between barrier wind and NE bora wind needs to be made on several IOPs in NEI region. Please discuss this explicitly in text, and list IOPs which satisfy conditions for barrier wind without any signs of northeasterly bora winds resulting from flow splitting over the Alps. Please discuss whether early phase of bora can reinforce barrier winds which are typically preceding bora onset? Please discuss this also in context of situation when blocked flow situation persists, resulting in low-level convergence well upstream of the orography. The interaction of southeasterly and northeasterly bora flow in the northern Adriatic is known to stimulate convection, so demarkation between barrier winds and bora is necessary.*

Barrier wind development over NEI area requires a southeasterly upstream flow over the Adriatic Sea that is deflected westwards over the northeastern part of the Po Valley. This situation is typically produced by the presence of a cyclonic circulation over the western Mediterranean, close to northern Italy or over the Genoa Gulf. However, once this configuration is attained, it is required that the low-level flow impinging on the Alps is stably stratified and that a quasi-stationary state is reached to prevent a rapid evolution (and dissipation) of the barrier wind.

Quite often (as in IOP18, see also Buzzi et al, 2020b and their Fig. 2), most frequently from mid-October to mid-April, the synoptic situation comprises an upper-level trough driving a strong south-southwesterly flow over the Adriatic Sea and northern Italy, while the barrier wind is confined below 850 hPa. Therefore, there is no northwesterly (cold) airflow impinging on and splitting around the Alps, so that the conditions for Bora are not attained. This is well described in Buzzi et al. (2020b), especially their Figs. 2, 5 and 6.

Only in some events (e.g., Davolio et al., 2009, Fig. 2a), colder northerly airflow may reach and cross the Eastern Alps, merging with the barrier flow, thus reinforcing the north-easterly wind over NEI; however, this configuration was not observed during any HyMeX IOP. In any case, this happens after the barrier wind is already well-established, thus Bora seems to play only a marginal role, as also stressed in Monai et al. (2006). (They also stress in their conclusions that "northeasterly flow over Veneto is not part of a classical Bora configuration".) In fact, this does not impact the blocked flow condition and the horizontal gradient of temperature and pressure already established by the deceleration of the impinging southerly wind (see also Markowski and Richardson, 2010), the latter being the key feature. Moreover, moisture supply for convection and precipitation is provided by the southerly flow over the Adriatic Sea. Finally, it is the passage of the cold front moving eastward that interrupts the incoming flow and the condition for barrier wind development, so that Bora does not have the time to intensify significantly.

For these reasons, it does not seem relevant to include explicitly the northeasterly Bora type flow in the conceptual model we are proposing. However, the latter model does not explain all the heavy precipitation events over NEI, but only those associated with intense low-level southeasterly flow interacting with the Alps, possibly in the presence of a pre-existing cold pool over the plain.

Strong convergence between Bora and Sirocco, also associated with intense rainfall, pertains to a different category of events, as occurred for example on 12 November 2019 (Ferrarin et al., 2021), characterized by a more complex mesoscale evolution including a displacement of the cyclone more to the south, over the Tyrrhenian Sea.

That said, we add in the text a sentence mentioning the possibility to have an enhancement of the barrier wind due to Bora wind, including the reference to previous papers.

Bibliography:

Buzzi, A., Di Muzio, E., and Malguzzi, P.: Barrier winds in the Italian region and effects of moist processes. Bull. of Atmos. Sci. and Technol., 1, 59–90, https://doi.org/10.1007/s42865-020-00005-6, 2020b.

Davolio, S., Buzzi, A., and Malguzzi, P.: Orographic triggering of long-lived convection in three dimensions, Meteorol. Atmos. Phys., 103, 35–44, 2009.

Ferrarin, C., Bajo, M., Benetazzo, A., Cavaleri, L., Chiggiato, J., Davison, S., Davolio, S., Lionello, P., Orlic, M., and Umgiesser, G., 2021: Local and large-scale controls of the exceptional Venice floods of November 2019, Progress in Oceanography, 197, 102628, 2021.

Markowski, P. M. and Richardson, Y.: Mesoscale meteorology in Midlatitude, John Wiley & Sons, 407 pp., 2010.

Monai, M., Rossa, A., and Bonan, A.: Partitioning of snowy and rainy precipitation in a case of a north Adriatic frontal passage, Adv. Geosci., 7, 279–284, http://www.adv-geosci.net/7/279/2006, 2006.

*211: Please include definition of the Froude number. In classical definition (U/NH), nearly neutral N mentioned in the first part of the sentence would lead to high Froude number and not low values such as specified in the second part of the sentence. Please clarify.*

Sometimes the Froude number is defined as NH/U, and we followed this definition in the previous manuscript version. However, we agree that the classical definition is more common, and we changed the text accordingly.

*220: Specify explicitly for which flow you mean to have high Froude number*

Clarified.

*234: Figure 3. In text first type of event corresponds to event shown on Fig 3.b. Please harmonize.*

Text reversed.

*251: Please consider demarcation between low-level jet and low-level jet stream as synoptically forced LLJ is usually termed "low-level jet stream" (Stensrud 1996).*

A footnote was added.

*275: Figure 4, caption. Please provide more information on the "run"*

Added.

*365: Figure 7. Please improve the readability of the figure.*

This is the original version of the figure received from the main author of the paper from which it has been taken. We do not have the data for re-plotting this picture. Although we agree the text is not clearly readable, we think that the main features indicated in the text are recognizable in this figure.

*452: Please reformulate "internal area" to enhance text clarity.*

Changed.

*461: Please discuss what caused the sudden increase of IWV which occurs prior to HPC in CI*

An explanation has been added; now it reads "A sudden increase in IWV of about 10 kg/m$^2$ occurred 6-12 h before the intense events, which received moisture from different sources (the transport of humidity was mainly from the Atlantic and from the southern Mediterranean); in particular, the

increase in the low-level humidity content strongly impacted on stability and convection intensity (Khodayar et al., 2018)."